# Topical diclofenac vs placebo for the treatment of chronic Achilles tendinopathy: A randomized controlled clinical trial

Erin Bussin[1], Brian Cairns[2], Tommy Gerschman[3], Michael Fredericson[4], Jim Bovard[5], Alex Scott[6]*

1 Fortius Sports Medicine, Burnaby, British Columbia, Canada, 2 Faculty of Pharmaceutical Sciences, University of British Columbia, Vancouver, Canada, 3 Department of Pediatrics, University of British Colombia, Vancouver, Canada, 4 Department of Orthopaedic Surgery, Stanford University, Stanford, California, United States of America, 5 Department of Family Practice, University of British Columbia, Vancouver, Canada, 6 Department of Physical Therapy, University of British Columbia, Vancouver, Canada

* ascott@mail.ubc.ca

**Data Availability Statement:** All relevant data are within the manuscript and its Supporting Information files.

## Abstract

### Introduction

The application of topical diclofenac has been suggested as a possible treatment for Achilles tendinopathy. Our aim was to answer the question, is topical diclofenac more effective than placebo for the treatment of Achilles tendinopathy?.

### Methods

67 participants with persistent midportion or insertional Achilles tendinopathy were randomly assigned to receive a 4 week course of 10% topical diclofenac (n = 32) or placebo (n = 35). The *a priori* primary outcome measure was change in severity of Achilles tendinopathy (VISA-A score) at 4 and 12 weeks. Secondary outcome measures included numeric pain rating, and patient-reported change in symptoms using a 7 point scale, from substantially worse to substantially better. Pressure pain threshold (N) and transverse tendon stiffness (N/m) were measured over the site of maximum Achilles tendon pathology at baseline and 4 weeks.

### Results

There were no statistically or clinically significant differences between the diclofenac and placebo groups in any of the primary or secondary outcome measures at any timepoint. Average VISA-A score improved in both groups (p<0.0001), but the improvements were marginal: at 4 weeks, the improvements in VISA-A were 9 (SD 11) in the diclofenac group and 8 (SD 12) in the placebo group, and at 12 weeks the improvements were 9 (SD 16) and 11 (SD13) respectively–these average changes are smaller than the minimum clinically important difference of the VISA-A.

**Funding:** Scott F. Nadler PASSOR Musculoskeletal Research Grant - AS, MF, BC, JB (acknowledgements) http://foundationforpmr.org/research-grants-2/scott-f-nadler-passor-musculoskeletal-research-grant/ The funders played no role in the study design, data collection and analysis, decision to publish, or preparation of the manuscript.

**Competing interests:** The authors have declared that no competing interests exist.

## Conclusion

The regular application of topical diclofenac for Achilles tendinopathy over a 4 week period was not associated with superior clinical outcomes to that achieved with placebo.

## Introduction

Achilles tendinopathy (AT) is characterized by pain, swelling and impaired function in the Achilles tendon [1,2]. Most people with Achilles tendinopathy present with a combination of risk factors such as repetitive loading of the tendon and advancing age [3]. The pain from tendinopathy is usually localized to the Achilles tendon, and is thought to be primarily nociceptive in most cases [4], although there is evidence that central sensitization may occur [5]. Histopathological reports reveal a slight inflammatory and fibrinous reaction in Achilles tendinopathy located mainly in the paratendon [6], with pathology in the tendon proper consisting of collagen disorientation and disarray, increased glycosaminoglycan, and neurovascular proliferation [7].

The potential roles of many inflammatory pathways and substances in the development of Achilles tendinopathy are poorly understood, in part due to a dearth of data from the early stages of pathology. A recent study of patellar tendinopathy patients with early symptoms reported that the average mRNA levels of Substance P (SP) and Transforming growth factor beta (TGF-β) were higher in patellar tendon biopsies of those with early symptoms compared to those without symptoms [8]. SP, when administered peripherally, is typically painful, and its upregulation has been observed in several other tendinopathies both in the chronic stage (gluteal, rotator cuff, Achilles) [9–11], as well as during the development of pathology in a laboratory model of Achilles overuse [12]. Diclofenac is capable of decreasing SP expression, and it has been suggested that this action may contribute to diclofenac's analgesic properties [13]. In a small, randomized cross-over trial, we found that 10% topical diclofenac reduced Achilles tendon pain after 3 days of treatment in people with Achilles tendinopathy [14]. Thus, we hypothesized that a longer-term treatment may result in a quicker improvement in tendon pain, along with an improved return to function. Our primary objective was to conduct a randomized placebo-controlled trial of a 4-week course of 10% topical diclofenac, and determine if diclofenac improved the severity of Achilles tendinopathy pain and function after 4 weeks, or at 12-week follow up.

## Methods

### Study design

This was a parallel, randomized, double-blind, placebo-controlled trial conducted at a single location (Fortius Sports Medicine in Vancouver). The trial was registered at ISCRTN (13557412) in October 2018, slightly after recruitment began due to an oversight on the part of the Principal Investigator. There are no ongoing or related trials for this intervention which required registration. The study was approved by the Clinical Research Ethics Board of the University of British Columbia (H18-00181) in April 2018, and all participants provided written informed consent. There were no changes to the ethics-approved protocol following commencement of the trial.

### Participants

Participants were recruited by placing posters in sporting clubs, running groups, gyms, community centres, running trails, and medical clinics in the Greater Vancouver area. We

included male and female subjects aged 19 years and older, fluent in English, who were previously diagnosed with Achilles tendinopathy by a health care professional (imaging at the discretion of the treating physician) and demonstrating the following criteria–localized tendon pain and thickening, worsened with palpation and tendon loading activities, and no clinical suspicion of other diagnoses. They had to have symptoms for 3 months or more and a Victorian Institute of Sport Assessment–Achilles (VISA-A) score of less than 80 (indicating the presence of Achilles tendon-related impairment). According to our understanding of Canadian guidelines for ethical research, it is a requirement that participants not be denied a standard level of care, therefore we required that participants be currently or previously engaged in a rehabilitation program for their Achilles problem. Both unilateral or bilateral cases were included, with all treatment and evaluation assigned to the worse side (as indicated by the participant). We excluded those with: BMI greater than 30.0, previous Achilles tendon rupture, chronic pain syndrome, diabetes, hyperlipidemia, metabolic syndrome, systemic inflammatory diseases, gastric ulcers, kidney disease or unstable hypertension, symptomatic osteoarthritis of the spine or lower extremities. We also excluded participants who had already received corticosteroid injections in the Achilles region, who were taking oral non-steroidal anti-inflammatory medication regularly, who had been prescribed anticoagulants or fluroquinolones within the past 5 months, who had known allergies to diclofenac or placebo gel, or who were unable to give informed consent.

## Randomization and allocation concealment

The principle investigator delivered the drug and placebo supplies to the research assistant in pre-measured, labelled bags, each containing a premeasured 4-week supply of drug or placebo. The drug and placebo appeared identical, contained in translucent orange syringe tubes, and the bags were labeled with randomly assigned numbers. The supply bags were delivered in balanced blocks of 16 (n = 8 drug and n = 8 placebo per block). After enrolling a new participant and conducting baseline measures, the research assistant pulled a supply bag at random and noted down the label number. The research assistant was responsible for all aspects of coordinating the study and collecting the data.

## Interventions

Participants were instructed not to do any moderate or vigorous physical activity during the 72 hour period preceding Visit 1, and to refrain from taking any oral NSAIDS for 1 week (and no analgesics for 24hrs) prior. After enrolment, participants were supplied with either 10% diclofenac gel (Medisca, PLO Mediflo 30, an opaque gel with an off-white hue), or placebo (gel only). Participants were instructed to massage 1 g of the gel along the Achilles tendon, including the most painful area of the tendon, for 30–45 s three times a day at 8-hour intervals for 4 weeks. They were asked to complete a medication administration diary to confirm their compliance with the regimen, to document any medical or paramedical treatments received, and an exercise diary to record their exercise and physical activities. Participants were told about the potential side effects of topical diclofenac (gastrointestinal upset or pain, heartburn, headache, dizziness, drowsiness, rash) and instructed to contact the study coordinator if they thought they were experiencing a side effect. In the case of a side effect, the principal investigator discussed the case with the attending physician (TG) who made the decision to exclude the patient or not. Participants were advised to initiate no new treatments throughout the study, and to not make any changes to treatment they might already be receiving.

## Study visits

The study included 2 in-person visits (at 0 and 4 weeks), and three email check-ins (at weeks 1, 3, and 12). At week 0, the study coordinator conducted an eligibility screen, explained the study, obtained informed consent, administered baseline measures and written (paper) questionnaires, and provided the study diary and gel. Participant characteristics were recorded, including age, sex, height and weight (measured by researcher), duration of symptoms, nature of symptom onset (acute or gradual), physical activity level (number of days active for at least 30 minutes, and Tegner Activity Score). An overview of the study visits is provided in Table 1.

## Outcome measures

The outcome measures below were collected, as planned prior to the start of the trial:

**VISA-A.** The primary outcome for which the study was powered was the change from baseline in 4 and 12 week VISA-A score (Victorian Institute of Sports Assessment–Achilles). This is a patient-reported outcome measure which records current level of physical activity or sport participation, and the amount of pain with those activities and with particular movements. A tendon with no symptoms would score 100, with progressively lower scores indicating worse impairment [15]. This was filled out independently by participants on paper in person at 0 and 4 weeks, and via an email questionnaire at 12 weeks.

**Pain.** Participants used a numeric Pain Rating Scale (0–10) to rate their average pain during tendon loading activities over last week, and their current pain. Zero represented no pain, and 10 the worst pain imaginable. Paper questionnaires were administered at weeks 0 and 4; and verbal, text or email ratings on weeks 1 and 3.

**Pressure pain threshold.** An AlgoMed Algometer (Medoc, Ramat Yishai, Israel) was used to measure pressure pain threshold (PPT) over the most affected part of the Achilles tendon. The participant lay face down on a treatment bed. Pressure was applied at 30 kPa/s until the first onset of pain, at which point the participant pushed a button to register the PPT in Newtons. If the person had resting pain, they were told to push the button when the pain first began to increase.

**Tendon stiffness.** A hand-held dynamometer (MyotonPRO, Myoton AS, Estonia) was used to measure transverse tendon stiffness, exactly as described previously [16]. Participants lay face down with calf relaxed and foot hanging freely. The MyotonPRO probe tip was positioned perpendicular to Achilles tendon in line with the centre of the medial malleolus, preloaded to 0.18N, and measured three times to obtain the average value in Newtons/meter.

**Table 1. Timing of data collection.**

|  | Baseline (week 0) | Week 1 (phone or email) | Week 2 (phone or email) | Week 4 (in person) | Week 12 (email) |
|---|---|---|---|---|---|
| VISA-A (Pain and function) | x |  |  | x | x |
| Numeric pain rating scale | x | x | x | x |  |
| Pressure pain threshold | x |  |  | x |  |
| Tendon stiffness | x |  |  | x |  |
| Participant global rating of change |  |  |  | x |  |
| Drug compliance |  |  |  | x |  |
| Questionnaire: non-protocol treatments |  |  |  | x |  |
| Document side effects |  | x | x | x |  |
| Questionnaire: Blinding efficacy |  |  |  | x |  |

X = data point collection. VISA-A: Victorian Institute of Sport Assessment–Achilles. NPRS: Numeric pain rating scale.

**Global rating of change.** Patient-reported change in symptoms was assessed at 1, 3 and 4 weeks using a 5 point scale from 1 (a lot worse) to 5 (a lot better), with 3 being neutral (unchanged).

**Drug compliance.** The amount of drug used was calculated by comparing the doses recorded in the participants' diaries with the prescribed amount, and expressed as a percentage.

**Non-protocol treatments.** Participants were asked at week 4 about non-protocol treatments, side effects, and their guess as to treatment allocation (success of blinding). At week 12, participants were contacted by email to fill out an online questionnaire.

**Side effects.** One and 3 weeks after visit 0, the study coordinator called or emailed the participants to inquire about possible side effects. Prompting questions about possible symptoms were included, such as indigestion, gas, diarrhea, and stomach pain.

**Blinding success.** The participants and study coordinator were blinded to group allocation. After 4 weeks, participants were asked which treatment they thought they received and their response recorded. They were asked to rate how confident they were in their guess, from 1 (not confident) to 5 (very confident).

## Sample size calculation

We conducted a power analysis using G*power (v 3.1) for a repeat-measures ANOVA with two groups and three timepoints, including testing for a within-between group interaction. For the main outcome measure (VISA-A), we assumed an effect size for diclofenac of F = 0.25 (medium), $\alpha = 0.05$, $1-\beta = 0.95$, correlation among repeated measures of 0.65, and $\varepsilon = 1$. This yielded a sample size of 32 per group.

## Statistical analysis

Prior to analysis, data were inspected for normality with the Shapriro-Wilk test. Data were normally distributed with the exception of change in pressure pain threshold (right skewed, $p<0.01$). We conducted repeat measures ANOVA for VISA-A and pain with activity or current pain, a Mann-Whitney U test for the change in pressure pain threshold and a paired t-test for tendon stiffness using statistical software (VassarStats.net, accessed May 5 2020 and Jan 6 2021). Data are presented to 2 significant figures with the standard deviation (SD) in parentheses. There were no interim analyses. Four unplanned analyses were carried out. We would like to emphasize that these analyses are not meant as a fishing expedition or to in any way water down the primary analysis, but rather provide additional supplementary information not intended to alter the primary conclusions of the planned analysis: (1) Drug compliance in the two groups was compared using a Mann Whitney U test; (2) the proportions of individuals who did or did not engage in rehabilitation in the two treatment arms was compared with a Fisher's Exact test; (3) the outcome measures at baseline were compared between and men and women using Mann-Whitney-U tests for pain and t-tests for VISA-A, pressure pain threshold and tendon stiffness; and (4) a linear mixed effect model for VISA-A outcome was used with treatment (placebo vs drug), sex, age, time, and physical rehabilitation (engaged or didn't engage during study) as the main effects. The genesis of these analyses was from collaborative discussions among the co-authors attempts to provide additional insight into the study's results, particularly the lower-than-anticipated participation in rehabilitation, and the unexpected trend toward worse outcome measures in women. All available data for all patients who received their allocated treatment are presented regardless of whether their data set was complete, however, the repeat measures ANOVA and paired t-test analyses were only included for complete data sets. All patients were analyzed according to their allocated treatment. We did

not impute or replace any missing values. We calculated summary statistics for the other outcomes.

## Results

### Participants

Recruitment began in May 2018, and was completed in January 2020 when the recruitment target was met. Sixty-seven participants were allocated into 2 groups (Fig 1) that were demographically and clinically similar (Table 2: no significant differences). There were no significant differences in baseline measures between men and women (S1 Table).

### Primary outcome: VISA-A at 12 weeks

The primary analysis indicated that there was no difference in VISA-A score between the diclofenac and placebo groups at any timepoint (p = 0.47, Fig 2). VISA-A increased during the course of the study indicating improvement in symptoms in both groups (p<0.01), but the average magnitude of the change at 12 weeks was not clinically significant: 8.5 (16) in the diclofenac group and 11 (13) in the placebo group.

   The exploratory analysis (mixed linear model) confirmed that there was a significant effect of time (p<0.001), with the major part of the improvement happening between weeks 0 and 4 regardless of treatment assignation or age, and also indicated that women tended to have worse outcomes (10 VISA-A points lower relative to men, with standard error of 4.0, p = 0.013) regardless of treatment allocation (S1 Fig).

### Pain

There was no difference in pain among the diclofenac and placebo groups, either during tendon loading activity (p = 0.17), or at rest (p = 0.86) (Fig 3). Pain with activity declined (p<0.01) during the course of the study, but the magnitude of difference was not clinically significant: 0.97 (2.1) in the diclofenac group and 0.70 (1.8) in the placebo group. Similarly, pain at rest declined (p<0.0001), but the magnitude was not clinically significant: 0.79 (1.8) and 0.94 (2.1) respectively.

### Pressure pain threshold and tendon stiffness

There were no significant differences in the pain pressure threshold or tendon stiffness after 4 weeks of diclofenac treatment (Fig 4).

### Global rating of change

On a 5 point Likert scale of patient-rated symptom change, there was no significant difference between the diclofenac (3.6 (1.6)) and placebo group (3.6 (1.0)).

### Drug compliance

The diclofenac group reported receiving on average 91% of the prescribed dose, compared to 88% in the placebo group: this was not significantly different (p = 0.496).

### Non-protocol and concurrent treatments

No participants reported receiving non-protocol medical treatments. Surprisingly, only a minority of participants engaged in active rehabilitation or other physical treatments during the study despite this being the recommended treatment for chronic Achilles tendinopathy.

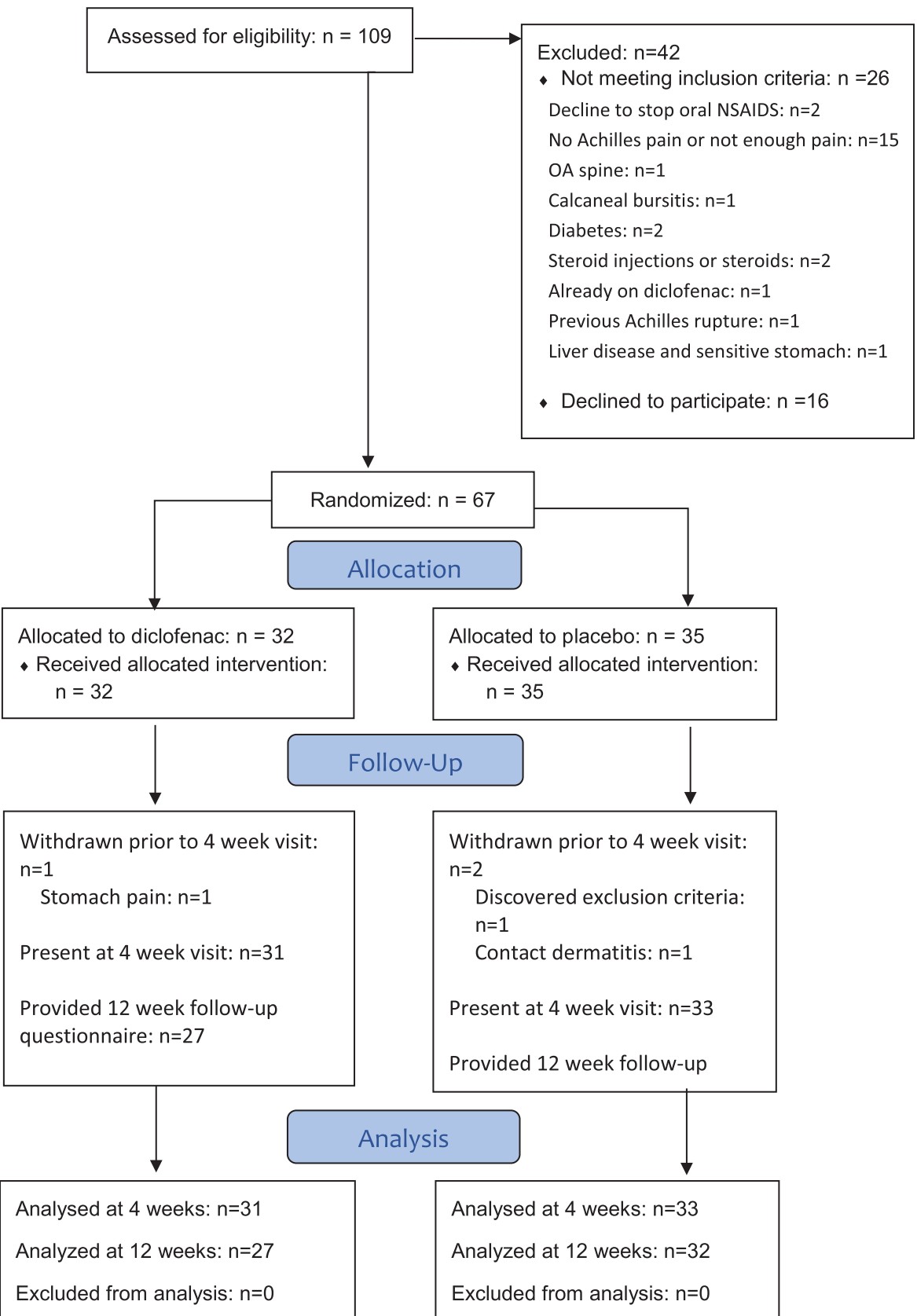

**Fig 1. Consort diagram.** The flow of all participants through the study from eligibility screening through to analysis is shown.

**Table 2. Participant clinical characteristics.**

| | Diclofenac (n = 32) | Placebo (n = 35) |
|---|---|---|
| **Age: mean (SD)** | 45 (11) | 45 (11) |
| **Sex: F, M** | 13 (41%), 19 (59%) | 18 (51%), 17 (49%) |
| **BMI, kg/m$^2$: mean (SD)** | 26 (2.9) | 25 (3.4) |
| **Height** | 175 (8.4) | 174 (10) |
| **Weight** | 79 (12) | 77 (14) |
| **Gradual onset (Y, N)** | 26 (81%), 6 (19%) | 29 (83%), 6 (17%) |
| **Location (M, I)** | 19 (59%), 13 (41%) | 25 (74%), 9 (26%) |
| **Symptom duration, months: mean (SD)** | 49 (71) | 65 (78) |
| **Tegner Activity Score** | 6.5 (1.4) | 7.1 (1.5) |
| **Days per week engaging in at least 30 min physical activity** | 4.3 (1.7) | 4.2 (1.3) |

SD–standard deviation. F–female. M–male. BMI–Body mass index. M–midportion. I–insertion.

Twelve diclofenac participants engaged in physical rehabilitation treatments: Nine physiotherapy and three other (massage, chiropractic, or strength conditioning). Seven placebo participants engaged in physical rehabilitation treatments: five physiotherapy and two other (massage, chiropractic, or strength conditioning). Exploratory analysis indicated that the effect of engaging in physical rehabilitation during the study did not lead to a statistically significant difference in VISA-A score (p = 0.66).

## Side effects

Thirteen participants in the diclofenac group, and 12 in the placebo group, reported mild gastrointestinal symptoms during the 4 week treatment period such as diarrhea, bloating/gas, nausea, heartburn, or stomach cramps. There were two reports of vomiting, both from participants in the diclofenac group: both participants attributed the symptoms to other causes (food

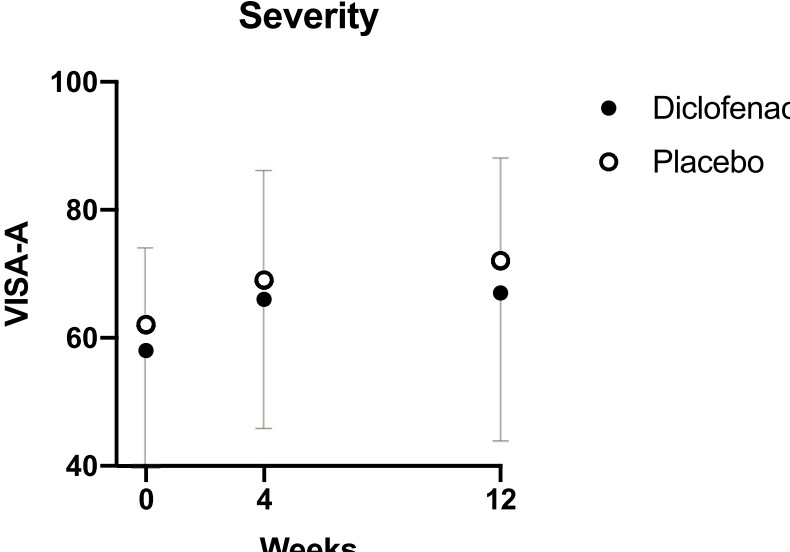

**Fig 2. Change in VISA-A over time.** VISA-A–Victorian Institute of Sports Assessment. The mean VISA-A score at each time-point is shown for the two treatment groups. Error bars represent the standard deviation.

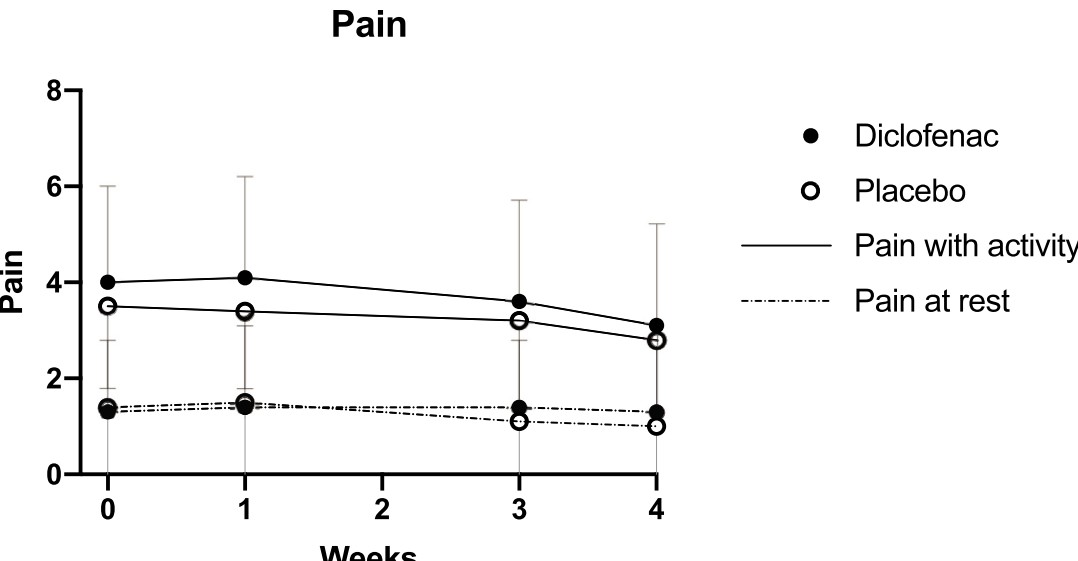

**Fig 3. Change in pain over time.** Mean pain ratings at each time-point are shown for the two treatment groups. Error bars represent the standard deviation.

poisoning, virus). One participant in the diclofenac group developed stomach pains and was unblinded and advised to withdraw from the study. Her stomach pain resolved spontaneously with no need for medical intervention. One participant in the placebo group developed contact dermatitis at the location of gel application, and was unblinded and instructed to withdraw from the study. The dermatitis resolved spontaneously without medical intervention once the gel application was discontinued. These two subjects are both included in Table 1.

## Success of blinding

For both groups, there was a tendency for participants to guess they were in the placebo group: in the diclofenac group, 11 guessed drug and 20 guessed placebo; and in the placebo group, 13

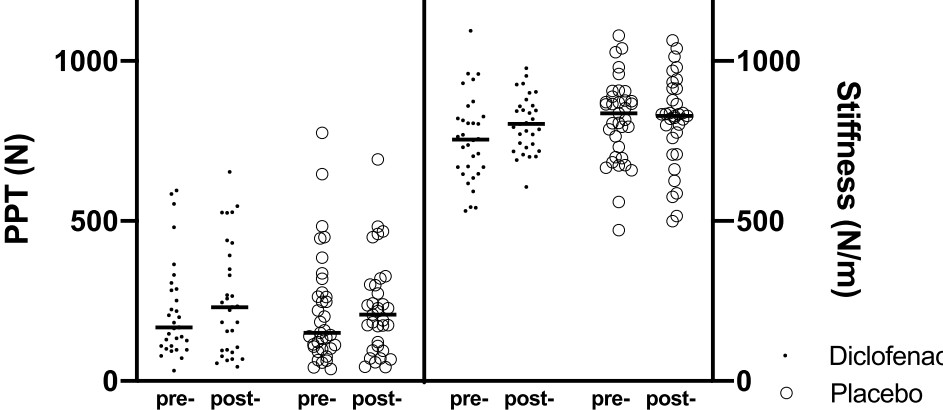

**Fig 4. Change in pressure pain threshold (PPT) and transverse tendon stiffness at 0 and 4 weeks.** "Pre-"denotes 0 weeks, "post-"denotes 4 weeks.

guessed drug and 20 guessed placebo. There was no difference in the average confidence of the guesses between the diclofenac group (mean 2.9) and placebo group (mean 3.0).

## Missing data

As seen in the Consort diagram (Fig 1), there were three withdrawals, and all data were missing from the point of withdrawal onward. As noted above, two withdrawals were due to medication side effects (one diclofenac and one placebo). The third withdrawal was an individual allocated to the placebo group who withdrew for personal reasons after the baseline visit. Aside from those missing values, the outcome measure data sets were complete with the following exceptions: 1 missing PPT value (placebo), 2 missing NPRS values (placebo), 16 missing drug compliance questionnaires (8 placebo, 8 NPRS), 1 missing concomitant treatment questionnaire (diclofenac), and 4 of the 12-week VISA-A scores (3 placebo, 1 diclofenac).

## Discussion

This study demonstrated that there is a lack of clinical response to 4 weeks of treatment with topical diclofenac compared with placebo, in people with Achilles tendinopathy, the majority of whom were not engaged in a supervised rehabilitation program during the course of the study. The participant sample was a relatively small group of individuals and included those with predominantly midportion, gradual-onset tendinopathy, with a smaller population who had insertional tendinopathy, distributed equally among the two treatment groups. The recruited sample were, on average, physically active and free from commonly associated diseases such as a metabolic disorder.

Previously, we reported that a 3 day course of diclofenac, delivered exactly as in the current study, resulted in a small amount of analgesia (average pain decrease from 4.8/10 to 3.1/10) [14], and wondered if a longer course of treatment would result in quicker functional improvements for people with Achilles tendinopathy. The current study prompts us to reject that notion, and conclude that the acute analgesia of diclofenac does not translate, by itself, into improved function in the majority of individuals. It is possible that if diclofenac were prescribed for pain control in participants who are also prescribed a program of therapeutic exercise, the results may have been different. We did not set out to compare those who did or did not engage in rehabilitation within the diclofenac group, but it is interesting to note that the average change in VISA-A at 12 weeks was 12 (12) in the rehab group and 6.9 (17) in the no-rehab group. In addition, it is interesting to note in the data set that in participants who received diclofenac that started out with a relatively high activity pain rating ($\geq$6, n = 7), the mean reduction in pain was 2.3, compared to 0.60 in those with pain scores of 5 or less (n = 24). Perhaps a future study could examine whether prescription of diclofenac specifically to those with high initial pain ratings would be a more effective approach.

There is very little research into the possible effects of anti-inflammatory strategies for Achilles tendinopathy, in contrast with other tendon-related conditions such as lateral elbow tendinopathy or rotator cuff-related pain. Anti-inflammatory treatments which have been examined include piroxicam, standard corticosteroid injection, and high-volume injection of saline (HVI) with corticosteroid and local anaesthetic (the effects of which are attributed primarily to the high saline volume). In a single mid-sized randomized controlled trial, participants treated with piroxicam [17] fared no better than those treated with placebo: both treatment groups also received a therapeutic exercise program, and the reported success rate was about 50% in both groups. DaCruz [18] et al found no significant response to treatment in either arm of an RCT in which 28 people with Achilles tendinopathy were randomized to receive a single injection of methylprednisolone acetate or placebo (with both groups also

receiving ice and therapeutic ultrasound): in this study, the reported response rate was 33%. A 2015 Cochrane review on injection therapies for Achilles tendinopathy concluded that the quality of evidence was of too low to base a recommendation for or against corticosteroid or other injections [19]. High volume injection of saline into the Achilles paratendon has shown promise in initial studies [20,21].

We were initially surprised by the lack of clinically significant improvement in the current study. Rompe et al reported that the VISA-A score increased from 48 to 55 in those allocated to a "wait and see" approach after 4 months, as opposed to from 51 to 76 in a group receiving therapeutic exercise: [22] the magnitude of improvement in the "wait and see" arm is similar to what was observed in both treatment arms of our study, which did not include a supervised rehabilitation program. One of the inclusion criteria was that participants had to already have engaged in, or be currently engaged in, a rehabilitation program; we required this because we felt it was unethical to enroll patients who had not received a basic level of care for their condition. However, because assessing the effects of rehabilitation was not our primary goal, we did not document the details of, or compliance with, rehabilitation, and surprisingly only a minority of participants reported continuing with their rehabilitation programs during this study. Possibly, enrolling in this study and the expectation of benefit from the diclofenac cream acted to disincentivize participants from persisting with rehabilitation, but this is purely speculative. Estimates of the MCID of the VISA-A score have ranged from 10 to 20, with the majority of studies using a value of at least 12 [discussed in 23]. In our diclofenac group, after 12 weeks only, 11 of 28 (40%) participants experienced a VISA-A increase of 10 or greater, which is similar to the placebo group (13 out 32, 41%). Perhaps the lack of clinical improvement in the majority of people in the current study was due to the lack of engagement in an exercise-based treatment for Achilles tendinopathy by the majority of participants. It should also be noted that the recording of VISA-A was done in person (on paper) at weeks 0 and 4, but by email at week 12: this variation in VISA-A measurement may have influenced the results.

The data collected indicate some potential sex-related differences (S1 Table, S1 Fig). Compared with men, VISA-A scores in women had a more modest increase over the treatment period. In men, VISA-A scores increased by 18% in the diclofenac group and 23% in the placebo group after 12 weeks, whereas in women, they increased by 13% and 11% at 12 weeks, respectively. In a study of painful eccentric exercise treatment of Achilles tendinopathy in men and women, it was found that the VISA-A score improved significantly more in men (27%) than in women (20%) at 12 weeks post treatment initiation [24]. This study reported that women over 50 had less pain improvement and were less satisfied with treatment than younger women. It is possible that these results represent a generally poorer response of women with Achilles tendinopathy to these treatments. In the present study, there was also a slight decline in VISA-A score at week 12 compared to week 4 in women who received topical diclofenac. This decline in VISA-A score was not observed in men in either treatment group, or in women who received the placebo topical therapy. Indeed, it has been recently reported that female sex is a significant factor in predicting treatment failure of corticosteroid injections for de Quervain's tenosynovitis [25]. Given these findings, it appears that treatment of Achilles tendinopathy in women with topical diclofenac should be avoided for most individuals, as it is unlikely to be beneficial and may even be detrimental to recovery.

A limitation of this study is that we did not record information on the use of loading tests or diagnostic imaging, or whether there was prior history (e.g. previous episodes) of tendinopathy at the same or other locations, as suggested by new guidelines [26]. A newly released consensus statement has indicated that tendinopathy clinical trials should include outcome measures from each of 9 domains [27], and we neglected to measure 2 of the domains (psychological factors and quality of life.

While inflammatory pathways such as local production of substance P are thought to contribute to the changes observed in tendinopathy [3,12], a number of structural changes occur in tendinopathy which a superficially delivered NSAID may be unable to counter, such as collagen disorganization and disarray, increased glycosaminoglycan, and neurovascular proliferation [7]. Perhaps a combination of exercise and other adjunct treatments aimed at restoring normal tissue function may be more successful at reducing nociception from the Achilles tendon and adjacent structures.

In conclusion, this study demonstrates a lack of clinical response to 4 weeks of treatment with either diclofenac or placebo, in healthy people with Achilles tendinopathy and no comorbidities. Expectations for people with Achilles tendinopathy who may be using diclofenac for analgesia should be based accordingly, and they should be encouraged to persevere with proven treatments such as exercise-based rehabilitation.

## Supporting information

**S1 Fig. Comparison of VISA-A in men and women.**
(EPS)

**S1 Table. Baseline comparisons across sex groups.**
(DOCX)

**S2 Table. Minimal data set.**
(XLSX)

**S1 File. Study protocol.**
(PDF)

**S2 File. Consort checklist.**
(DOC)

## Acknowledgments

The authors would like to thank Dr. Jason Crookham for engaging openly in the initial discussions about this project.

## Author Contributions

**Conceptualization:** Brian Cairns, Jim Bovard, Alex Scott.

**Data curation:** Alex Scott.

**Formal analysis:** Brian Cairns, Alex Scott.

**Funding acquisition:** Brian Cairns, Michael Fredericson, Jim Bovard, Alex Scott.

**Investigation:** Erin Bussin, Tommy Gerschman, Alex Scott.

**Methodology:** Michael Fredericson, Alex Scott.

**Project administration:** Erin Bussin, Alex Scott.

**Resources:** Erin Bussin.

**Software:** Brian Cairns.

**Supervision:** Brian Cairns, Tommy Gerschman, Alex Scott.

**Visualization:** Brian Cairns, Alex Scott.

**Writing – original draft:** Brian Cairns, Alex Scott.

**Writing – review & editing:** Erin Bussin, Brian Cairns, Tommy Gerschman, Michael Fredericson, Jim Bovard, Alex Scott.

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
