## [Decision Letter · Decision Letter 0]

6 Jan 2021

PONE-D-20-23822

Topical Diclofenac vs Placebo for the Treatment of Chronic Achilles Tendinopathy: A Randomized Controlled Clinical Trial

PLOS ONE

Dear Dr. Scott,

Thank you for submitting your manuscript to PLOS ONE. After careful consideration, we feel that it has merit but does not fully meet PLOS ONE’s publication criteria as it currently stands. Therefore, we invite you to submit a revised version of the manuscript that addresses the points raised during the review process.

Please be sure to address point-by-point all of the reviewer concerns and indicate your changes to the manuscript. Please ensure that your manuscript adheres to PLOS ONE’s publication criteria.

We look forward to receiving your revised manuscript.

Kind regards,

Johannes Fleckenstein

Academic Editor

PLOS ONE

Journal Requirements:

2. Thank you for submitting your clinical trial to PLOS ONE and for providing the name of the registry and the registration number. The information in the registry entry suggests that your trial was registered after patient recruitment began. PLOS ONE strongly encourages authors to register all trials before recruiting the first participant in a study.

1) your reasons for your delay in registering this study (after enrolment of participants started);

2) confirmation that all related trials are registered by stating: “The authors confirm that all ongoing and related trials for this drug/intervention are registered”.

Please also ensure you report the date at which the ethics committee approved the study as well as the complete date range for patient recruitment and follow-up in the Methods section of your manuscript.

Reviewers' comments:

Reviewer's Responses to Questions

**Comments to the Author**

1. Is the manuscript technically sound, and do the data support the conclusions?

Reviewer #1: Yes

Reviewer #2: Partly

Reviewer #3: Yes

2. Has the statistical analysis been performed appropriately and rigorously? 

Reviewer #1: Yes

Reviewer #2: Yes

Reviewer #3: Yes

3. Have the authors made all data underlying the findings in their manuscript fully available?

Reviewer #1: Yes

Reviewer #2: Yes

Reviewer #3: Yes

4. Is the manuscript presented in an intelligible fashion and written in standard English?

Reviewer #1: Yes

Reviewer #2: Yes

Reviewer #3: Yes

5. Review Comments to the Author

Reviewer #1: A two-arm randomized clinical trial was conducted to test the effectiveness of topical diclofenac. Effectiveness was measured as change in achilles tendinopathy (VISA-A score) at 4 and 12 weeks. No statistically or clinically significant differences were observed between the treatment and placebo arms for any of the outcomes.

Minor revisions:

1- Indicate if the distribution of the data was checked for normality prior to applying the ANOVA or t-tests.

2- Table 2: Indicate the percentages which correspond to the frequencies in Table 2.

Reviewer #2: Thank you for the opportunity to review this paper. On the whole it is well written, but I think there are some further points for consideration. Please see below.

The participants are not representative of the wider Achilles tendinopathy population; they are recruited from running clubs and sports clubs and were excluded if they had any commonly associated disease, such as a metabolic disorder. Not surprisingly they were all very active with low pain scores and high VISA-A scores. I think the title would benefit from changing to reflect this. This should also be considered in the discussion and conclusion.

The reporting of participant characteristics should be in line with recent recommendations: https://bjsm.bmj.com/content/54/11/627. Missing characteristics should be considered in the limitations section

Method used to generate random allocation sequence would benefit from justification. Why was this process used and not a more usual approach e.g block randomisation.

How sample size was determined would benefit from justification. Why a repeated Anova is used when the primary outcome is between group differences is not clear. This could leave the study under-powered and thus not able to detect a difference – 32 per group is a small number.

Clinical trials should include a measure for each of the nine core domains at a minimum, so that future meta-analyses will be able to better estimate treatment effects. https://bjsm.bmj.com/content/54/8/444. Missing domains should be discussed and considered in the limitations section

The clinometric limitations of the VISA-A should be discussed. https://journals.sagepub.com/doi/pdf/10.1177/1071100718816953?casa_token=QBSv-wQQ9h8AAAAA:I5KUyJK4cGnFG-CQytrhvpbc4faxF2gHs1qg2j8aobFpGmhIIQ16HO8y5uDERpaqBrItufDt0K7KoyI

https://bjsm.bmj.com/content/52/19/1221.abstract

Given the high levels of scoring, could the tool be no be sensitive enough to change?

Line 145 is repeated and should be deleted

There are two full stops at the end of the last sentence on p.6.

Reviewer #3: Thank you for the opportunity to review this interesting and important paper examining the efficacy of topical diclofenac in the management of Achilles tendinopathy (AT). The paper has been well-written and has simple but solid methods, appropriate to answer the research question. Whilst I have a few comments to make, most are minor and/or cosmetic, which I expect the authors to be able to address with relative ease. Likewise, I trust the authors will receive these comments with the constructive and collegial nature of their intention.

INTRODUCTION: No comments

METHODS:

L90. One inclusion criterion is that the patient must be "currently or previously engaged in a rehabilitation programme". I don't see the relevance of this to the research question. Furthermore, as you point out in the discussion, the nature of the involvement (or not) in a rehabilitation programme served as a confounding factor for the findings. I think the paper would benefit from a clearer justification for this issue, as well as further commentary in the discussion.

L120. Concerning the "potential side-effects of diclofenac", as a non-medical practitioner I would value a summary of the key side-effects that might be relevant to this drug and as a science-reader, would like to know which side-effects the participants were informed of and by what basis participants were excluded should they occur (as was the case on occasion).

L133. Table 1; NPRS is included within the Table title. I think it belongs as a table footnote (where you have placed the VISA-A abbreviation (L134)). Furthermore, you haven't provided an expansion for the abbreviation NPRS. This should be rectified.

L137. I don't understand this statement. We are in the methods section, and I read this as you describing results, yet with no detail. Please address this to allay confusion.

L145. The VISA-A was completed in person and then via email at a later date. My understanding of the VISA-A literature is that its validity across these different media has not been established. This may seem a little pedantic but in addition to that, my personal experience in executing this questionnaire leads me to believe that it might be possible that potentially significant sampling error may be introduced by this. I don't know if it would be sufficient to change the overall study findings, but I feel compelled to point this out and recommend you mention it in the discussion.

L145. This sentence effectively repeats the sentiment of Lines 142-4. Suggest removal.

L151. As per L145.

L194. I am curious regarding the genesis of the "unplanned analyses". Can you please provide a more thorough justification for this? As you describe and then discuss these findings they seem reasonable, but in order not to appear as though you were fishing, the nature of unplanned/post-hoc analyses should be described with more clarity.

RESULTS:

L213. Table 2; can you confirm that there was no difference in symptom duration between the diclofenac and placebo groups?

L222 (& L239, L249, L250, L256). I would like to see evidence from you statistical testing articulated here (eg p values). It would be best approached I think as a Table.

L249. "...non-significant trend for PPT to be higher at 4 weeks." Given significance testing (for all its faults), provides a binary outcome. I don't think it is appropriate to articulate such findings in this manner; it speaks of bias. Please remove.

DISCUSSION:

L305. In your opening discussion paragraph I think value would be added by summarising your key findings, including p-values.

L338. You describe the Fredberg study, but have not provided any relevance to the inclusion of it in your discussion. ie you do not allude to their findings and certainly do not describe them in the context of your study. Recommend revising this or removing altogether.

L358. Can you provide reference(s) for the estimates of MCID. To my knowledge there is less validation of MCID than you imply here. Please clarify.

6. PLOS authors have the option to publish the peer review history of their article (what does this mean?). If published, this will include your full peer review and any attached files.

Reviewer #1: No

Reviewer #2: No

Reviewer #3: **Yes: **James Debenham

---

## [Author Response · Author response to Decision Letter 0]

7 Jan 2021

Please see attached response to reviewers, and many thanks for the opportunity to revise and improve the manuscript.

We have also answered the editor's query about the trial registration.

---

## [Decision Letter · Decision Letter 1]

11 Feb 2021

Topical Diclofenac vs Placebo for the Treatment of Chronic Achilles Tendinopathy: A Randomized Controlled Clinical Trial

PONE-D-20-23822R1

Dear Dr. Scott,

We’re pleased to inform you that your manuscript has been judged scientifically suitable for publication and will be formally accepted for publication once it meets all outstanding technical requirements.

Kind regards,

Johannes Fleckenstein

Academic Editor

PLOS ONE

Additional Editor Comments (optional):

Reviewers' comments:

Reviewer's Responses to Questions

**Comments to the Author**

1. If the authors have adequately addressed your comments raised in a previous round of review and you feel that this manuscript is now acceptable for publication, you may indicate that here to bypass the “Comments to the Author” section, enter your conflict of interest statement in the “Confidential to Editor” section, and submit your "Accept" recommendation.

Reviewer #1: All comments have been addressed

Reviewer #2: All comments have been addressed

Reviewer #3: All comments have been addressed

2. Is the manuscript technically sound, and do the data support the conclusions?

Reviewer #1: (No Response)

Reviewer #2: Yes

Reviewer #3: Yes

3. Has the statistical analysis been performed appropriately and rigorously? 

Reviewer #1: (No Response)

Reviewer #2: Yes

Reviewer #3: Yes

4. Have the authors made all data underlying the findings in their manuscript fully available?

Reviewer #1: (No Response)

Reviewer #2: Yes

Reviewer #3: Yes

5. Is the manuscript presented in an intelligible fashion and written in standard English?

Reviewer #1: (No Response)

Reviewer #2: Yes

Reviewer #3: Yes

6. Review Comments to the Author

Reviewer #1: (No Response)

Reviewer #2: (No Response)

Reviewer #3: All of the comments from my review have been attended to and meet my satisfaction. Thanks and well done.

7. PLOS authors have the option to publish the peer review history of their article (what does this mean?). If published, this will include your full peer review and any attached files.

Reviewer #1: No

Reviewer #2: **Yes: **Adrian Mallows

Reviewer #3: **Yes: **James Debenham

---

## [Editor Report · Acceptance letter]

16 Feb 2021

PONE-D-20-23822R1 

Topical Diclofenac vs Placebo for the Treatment of Chronic Achilles Tendinopathy: A Randomized Controlled Clinical Trial 

Dear Dr. Scott:

I'm pleased to inform you that your manuscript has been deemed suitable for publication in PLOS ONE. Congratulations! Your manuscript is now with our production department. 

Kind regards, 

on behalf of

Priv.-Doz. Dr. Johannes Fleckenstein 

Academic Editor

PLOS ONE